# Health Consequences of Intensive E-Gaming: A Systematic Review

**DOI:** 10.3390/ijerph20031968

**Published:** 2023-01-20

**Authors:** Yinhao Shen, Antonio Cicchella

**Affiliations:** 1International College of Football, Tongji University, Shanghai 200092, China; 2Department for Quality-of-Life Studies, University of Bologna, 47921 Rimini, Italy

**Keywords:** professional Esports players, heart, sudden death, blood pressure, stress, heart rate, seizures, posture

## Abstract

The aim of this review is to examine the links among the different factors that determine harmful or even deadly events in professional and semiprofessional intensive Esports players. Cases of serious injuries or even death in young (<35 years old) male professional Esports players are reported every year. Fatalities and injuries in professional Esports players (PEGS) have only affected male players, and these events have mostly been concentrated in Asia. Studies in the literature have reported several causes and mechanisms of injuries. Links between injuries and previous comorbidities have emerged from the extant literature; obesity and/or metabolic disorders, seizures (associated with overstimulation of the eyes), heart malfunctions, high basal and abrupt increases in systolic blood pressure (SBP), prolonged stress, and poor posture have been associated with injuries. Several clinical signs have been identified and the question emerges whether or not self-regulation by Esports associations or public health authorities is necessary.

## 1. Introduction

Professionalism in Esports is relatively new, and the number of professional and semiprofessional Esports players (PEGs) is growing quickly [1]. PEGs are mostly concentrated in Asia, especially in China [1]. A large survey on Esports engagement that involved 27,454 respondents found that 47% in China Mainland, 40% in Indonesia, 39% in Thailand, 38% in Taiwan, 36% in Philippines, 30% in Vietnam, and 29% of respondents in Malaysia were engaged in Esports. In Europe, the countries with the highest number of Esports players were Spain (17%) and Italy (13%) [1]. On average, PEGs show a daily engagement of up to 15 h/day [2] and perform an average of 6.6 moves per second and up to 400 moves per minute [2].

The intensity of effort in Esports is linked to the type of game (e.g., violent vs. nonviolent) and to the rules of the tournament [3]. In 2022, Esports were included in the Asian games as a medal event [4].

Among PEGs, the weekly reported playing hours are 33.3 h as compared with 22.8 h among nonprofessional but competitive e-gamers [3]. In a study of 2867 PEGs, sampled from a total cohort of 136,920 e-gamers, only 6.74% (199) of the e-gamers were females [5]. This lower number of female players can explain the low rate of deaths among female Esport players. However, no epidemiological studies have been found about this issue. Contrary to “real” sport tournaments that were halted by the pandemic, competitive video gaming was not affected by the lockdowns [3].

Clinical guidelines have been developed for a safe approach to gaming to prevent and to treat the major physical and psychological illnesses caused by Esports [3,6]. These recommendations have mainly focused on the musculoskeletal, cognitive, vision, sleep, and nutritional aspects. Deaths have been reported from Esports tournaments and even from the first video game tournaments as early as the 1980s [7,8]. The economics of Esports is growing and involves the insurance sector for injury compensation [9]. A performance model and the mechanisms of cardiovascular and stroke injuries in Esports have been poorly investigated because of difficulties in obtaining live physiological data during high-level tournaments, without interfering with the game [10]. In 2013, the American Psychological Association (APA) DSM-5 manual included “gaming disorders” in the section on “Conditions for Further Study” [11] and, later in 2019, the World Health Organization (WHO)’s International Classification of Diseases (ICD-11) added “gaming disorders” to the list of recognized addictive disorders. However, psychiatric and psychological diseases seem to just marginally affect professional Esports players and, thus, have not been considered in this review, except for those related to physical injuries.

Contrary to traditional Olympic sports (except for extreme sports) that have been adapted to the physiological capacities of the players by fixing games with limited durations, Esports present an indefinite duration and the intensity of effort depends on several uncontrolled factors. These facts pose a serious concern for the health of the Esports players. Because the physiological causes of fatalities in Esports are still unclear, the aim of this paper is to review the available literature on possible physiological causes underlying injuries (including deaths for cardiovascular causes and stroke) in Esports intensive e-gamers.

## 2. Methods

A survey of the existing literature was conducted in four different databases from the inception to May 2022 (PubMed, Sport Discus, Ebsco, and Psychinfo) and on Google Scholar. The following keywords were used in different combinations: “injuries in Esports”, “medical aspects of Esports”, “professional e-gamers”, “stroke and e-gaming”, “heart attack and e-gaming”, “sudden death and Esports”, “pathophysiology of sudden death”, and “heart failure and Esports”. The search strategies were combined, and duplicates were removed using Endnote X7 (Clarivate Analytics, previously Thomson Reuters, Philadelphia, PA, USA). The databases were queried in a hierarchical way (e.g., first the broader database), starting from Google Scholar. All titles and abstracts were carefully read, and relevant articles were retrieved for review. In addition, the reference lists from both original and review articles retrieved were also reviewed. The eligibility criteria limited the search to studies performed on PEGs (not occasional), and to studies in which clinical, biochemical, neurological, and instrumental signs were related to heart attack, stroke, and serious injuries. To distinguish between PEGs and non-PEGs, we used the criteria adopted by a previous study [5], i.e., self-reporting to be a PEG, participating in professional tournaments, and living on an income from Esports. The papers were selected in the English language and psychological studies were excluded (except for those dealing with the psycho-/physiological factors that could influence the heart or brain’s physiological health, for example, sleep). The review complied with the PRISMA statement for a systematic review [12]. The inclusion criteria were: studies related to PEG deaths, fatalities, or injuries; experimental studies (physiological or questionnaires); case studies; or reviews. The exclusion criteria were: (i) studies written in languages other than English, (ii) studies involving non-PEGs, (iii) Congress abstracts. No limits were set concerning the year of publication. The inclusion or exclusion of articles was determined by applying the above criteria on the title and abstract as a first screening and on full texts as a second screening.

The review was registered in the PROSPERO database with registration no. 324822.

## 3. Results

In total, 56 papers met the eligibility criteria and were included in the review. Among the retrieved studies, 11 were cases studies, 10 were epidemiological, 12 were reviews, and 23 were experimental investigations (Figure 1). Among the retrieved papers, only nine studies were performed with professional Esports players and are reported in Table 1.

Table 1 provides a summary of the studies considering only professional and semiprofessional e-gamers.

Physical stressors are the most studied causes of fatalities in Esports [3]. As reported in the literature, most deaths are among males [3,5,6], which does not seem to reflect the gender distribution of e-gamers, since, in the age range 21–35 years, 20% of e-gamers are male and, in the general e-gamers population, 15% of e-gamers are male [21]. The term “stress” is broad, and in the psychological field, stress relates to disordered gaming (DG). DG has been defined by the American Psychological Association (APA) as “excessive and harmful use of video games, both online and offline, leading to significant functional impairment and/or clinical (psychological) distress” [10]. In a sample of 5734 professional Esports players, psychological distress was reported by only 14.3% of PEGs, and therefore, overall, the players did not perceive themselves as stressed [5]. In physiological sciences, the current accepted definition of stress concerns the response of the hypothalamic–pituitary–adrenal (HPA) axis and activation of the autonomic nervous system [22]. This activation is physically measurable, and it is associated with secretion of cortical hormones (mainly cortisol). Sudden and massive activation of the HPA axis results in increased occurrence of strokes and heart problems [22]. Activation of the HPA axis above normal levels has been observed at rest in subjects with depression and anxiety, symptoms which, in turn, are worsened by social isolation [5,23]. Chronic systemic pathological conditions of the vascular system are mostly found in PEG subjects who are younger than 35 years old as compared with those older than 35 years old [24]. Stroke is a thromboembolic event which targets large and small vessels in the brain [5,25,26] and is often associated with pre-existing chronic systemic conditions such as general inflammation and cardiometabolic diseases, and can depend on previous thrombosis [24,27]. Lower limb thrombosis is quite common in Esports players, to the point of minting the term “e-thrombosis” [27]. Thrombosis, however, can also be fatal in young subjects [28]. Adding to the risk factors for fatalities, it has been observed that seizures are also common in heavy video gaming [13,14] due to optical stimulation caused by the flashing lights of the games, especially blue lights. The incidence of seizures caused by visual stimulation has been reported to be 1/4000 in observed populations of nonprofessional Esports players; in PEGs, there have been no reported cases of seizures [29], albeit a higher incidence of seizures in young Esports players, for example, 352 cases per year in four European countries were reported in a 2002 study [30]. Seizures must not be underestimated because they can be a causal factor of stroke [14,29]. Long sitting hours are well known to be associated with obesity, and increased obesity rate and obesity-related hypertension are expected to increase the incidence of fatal strokes in young people [31,32]. In some Asian countries such as China, the diffusion of smoking habits in Esports players has worsened the predisposing factors to stroke [33]. An acute triggering factor for stroke and heart attack in PEGs is the consumption of neuroactive, capacity-enhancer stimulants during gaming, such as caffeine [15] and other stimulants [34]. In a study of 50 PEGs during an international competition in China (mean age 20 ± 1.67 years, BMI 22.86 ± 2.11), 11.22% of the PEGs were smokers, 19.38% were defined as usual alcohol consumers, and 25.86% were regular users of caffeinated beverages [35]. This percent of caffeine users seems to be especially high, because in Asia, coffee is not a common beverage among young people. In the same study, 12% of the sample reported symptoms of angina pectoris.

Myocardial infarction is defined as a syndrome where “myocardial cells die due to imbalances between myocardial oxygen supply and demand” [36]. It can be secondary to acute atherosclerotic plaque disruption (type 1 myocardial infarction) or alterations in myocardial oxygen supply and/or demand in the absence of acute atherothrombosis (type 2 myocardial infarction) [36].

In PEGs, cardiac function, HR, and heart rate variability (HRV) have been extensively investigated. Cardiac coherence, defined as the synchronization of the rhythm of breathing to the rhythm of the heart pulse, has been shown to be disrupted by intensive gaming because of the high level of stress [37]. Moreover, a strong association between hours of gaming and high SPB has been found in overweight adolescents [38]. Increased SPB and decreased HRV have been associated with fighting-type games in e-gamers, indicating a cardiovascular stress response [16,39,40]. In a study that investigated a more intensive gaming period in competitive young subjects (two males 15–19 years old) during a two-hour gaming session [41], statistically significant increments of both diastolic (DBP) and systolic (SBP) blood pressure were reported as compared with a matched control performing the same task. SBP measurements were 109 ± 11 vs. 122 ± 15 mm Hg and DBP measurements were 68 ± 8 vs. 78 ± 13 mm Hg [41].

Heart rate (HR) during e-game competitions can reach significantly high frequencies. The mean HR values during e-games have been measured at 120 ± 16 bpm, with peaks of up to 160–180 bpm [16,17,37,39,40], depending on the game. In collegiate Esports players (18 years old), the mean values were measured at 131.4 ± 19 bpm during a two-hour intensive session [42]. The RR intervals of the ECG showed significant alterations during a single gaming match; RR intervals were significantly lower during competition (465.71 ± 68.99 ms) as compared with pre-competition (643.64 ± 138.54 ms) [43]. It is worth noting that these values were observed during a game session of short duration and in a student population of e-gamers; it must be considered that PEGs in a competition usually play continuously for up to 6 h at high intensities [42]. Other studies have shown an increase in the activity of the nervous sympathetic system (increase in the low frequency/high frequency ratio of HRV) during heavy gaming [10,44]. During gaming in professional players, a new instrument for the detection of EEG, HR, and HR variability has been used to measure cardiac function; the instrument was based on light photodetectors that were minimally invasive in vivo [10]. The low frequency (0.04–0.15 Hz) and the high frequency (0.15–0.4 Hz) ratios of the RR intervals were used to indicate a balance between the activation of the sympathetic and the parasympathetic nervous system. During the beginning of a gaming session, the sympathetic nerves became more active, and the parasympathetic nerves became less active (LF/HF ratio = 3.56  ±  2.30 vs. 1.80  ±  1.06) in a group of 28 males and three females (mean age 25.79  ±  2.36 years old) [10]. In the same study, a decrease in the transit time of the pulse wave propagating from the heart to the periphery during the competition was observed (204.32  ±  25.78 vs. 84.94  ±  29.32 ms), meaning that the sympathetic nerves became more active. Unfortunately, even though the authors stated that a gaming session lasted 20 to 40 min (in this case, League of Legends was used as the test game) they recorded only the first 5 min of gaming, and no data under heavy fatigue were recorded.

Sleep cycle disruptions are also common in PEGs. Activation of the sympathetic nervous system, especially in evening hours, affects the sleep pattern [18]. In PEGs, the effects of gaming on sleep are larger, as shown in another study performed in South Korea [9]. In a multicentric study performed in South Korea, USA, and Australia, the results showed that South Korean Esports athletes had delayed sleep patterns, experienced prolonged wake periods after sleep onset, and slept less than 7 h per night [18]; these sleep patterns have been associated with mood swings (depression) in some Esports athletes [9]. South Korea has the largest numbers of registered Esports players, i.e., about 15 million people or roughly 30% of the population, and has a higher number of deaths among Esports players [45]. Therefore, South Korean PEGs have been extensively investigated. The USA, China, Thailand, and Taiwan have recorded several fatalities, mainly during e-gaming marathons, i.e., events often lasting overnight for 3 or 4 days [45]. In South Korea, recently, a case of “collapsing lungs” was reported due to a thromboembolic event during an Esports session [45]. A study investigating the neurophysiological aspects of e-games showed that e-gamers had faster reaction times than non-gamers, even higher than in professional basketball players [19]. This finding can be partially explained by the higher consumption of caffeine in Esports players [46]. Faster reaction times are signs of hyperactivity of the autonomic nervous system.

Death from heart injury can be caused by excessive accumulation of reactive oxygen species both locally in the heart [35] and in the gut [47], provoked by prolonged and intense aerobic exercise (oxidative stress). Increased oxidative stress can be an outcome of previous fatigue, sleep disruption, and autonomic system activation in Esports players [46]. The role of sleep deprivation in causing stress and ROS accumulation has been supported by several studies [48,49]. Air travel prior to competition, ingestion of caffeine and other stimulants, and stimulation by lights in the late evening hours or even overnight have been identified as the main factors for sleep disturbance, albeit no experimental data exist that link these factors to sleep disruption in Esports [48,50]. We did not find any EEG sleep studies on the sleep quality of PEGs (e.g., investigating sleep phases). A model of the interactions among fatigue, poor sleep, oxidative stress, and activation of the autonomic nervous system is shown in Figure 2.

A study on the Esports players’ eyes is of special interest, in which 18% of PEGs reported eye strains [50]. The eyes are of special concern because they can be damaged due to a sudden increase in BP and by vasoconstriction induced by gaze fixation [51]. The eye’s response to flashing lights is associated with overstimulation of the autonomic system and with possible onset of seizures. To understand the onset of seizures, it is interesting to study the areas of the brain involved in information processing during gaming. An fMRI study [52] showed that there were several brain areas active during an intensive gaming tournament: the left superior, medial, and middle frontal gyrus, which are involved, respectively, in working memory, spatial processes (included vision), monitoring and manipulation [53]. The left and right precuneus have been reported to be activated which are involved in body (and hand) representation, action prediction, and action selection [54]. In addition, the left angular post cingulate gyrus has been reported to be activated, which is related to impulsivity scales [55]. Other reported activated areas include the right pre-central gyrus, which is associated with the amplitude of movement perception [56], and the left-middle and left-superior temporal gyrus, which is related to the replication of hand gestures [57]. Lastly, the left postcentral (hand control [58]), right superior temporal (spatial processing [59] and speech perception [60]), and right middle temporal gyrus have been reported to be activated, which are also connected with hand manipulation and representation [61]. This localization is also be of special interest for the etiology of stroke; 32% of PEGs have reported frequent headaches and, even though headaches have not been directly associated with stroke, these are signs that should not be underestimated [51].

Considering heart failure, the primary causes of sudden cardiac death in individuals 14–35 years old (age of the mostly reported deaths) have been related to arrhythmogenic right ventricular cardiomyopathies [62] and lethal ventricular arrhythmia [62,63], consistent with the alterations observed in the heart of PEGs [64]. Mental stress induced by video gaming has been observed to increase the QT interval on an ECG [63,65]. An increase in the QT interval has been associated with sudden death [66,67,68] due to overstimulation of the heart sympathetic system, together with an associated sudden increase in BP [69].

Therefore, adrenergic stimulation associated with the powerful emotions of electronic gaming may trigger life-threatening arrhythmias. This phenomenon can happen in a small subset with underlying arrhythmogenic conditions [64] and can even occur in young subjects not known to have pre-existing cardiac conditions. Syncopes associated with emotional responses during violent video games are relatively common and should prompt a cardiac evaluation [65]. It has been observed [68] that mental stress and heightened emotions can shorten ventricular action potentials and trigger cardiac arrhythmia, particularly in long-QT syndrome [67,68]. An increase in sympathetic activation and in adrenergic stimulation have been proposed as the mechanisms for this phenomenon [63,66].

The main biochemical mediator of stress is cortisol, which has been found to be elevated in players who spent more than 5 h/day at a console [70]. Further, a case of rhabdomyolysis (damage to muscular myofibers with subsequent release of muscle material into the blood) was reported in a 40-year-old man who played strenuously for 3 days, and afterward, presented significant lesions in the upper arms and shoulder muscles [20]. Rhabdomyolysis can also be a cause for thrombosis due to increased viscosity of the blood.

## 4. Discussion

Severe and even deadly cardiovascular injuries are common among PEGs. The recent increase in the number of PEGs, fostered by social isolation, is now a public health concern because of the occurrence of fatalities during Esports. Heart attack and stroke in Esports are a paradigm of how different concurrent factors interact in causing a fatality.

Participation in professional e-gaming is growing. The number of PEGs and semi-PEGs increased significantly with the onset of the COVID-19 pandemic [10], and in the next years, a further increase is expected. The growing number of Esports players will lead to an increase in gaming-related health problems and fatalities. In the present review, we examined the causes of cardiac events in Esports, including sudden deaths due to heart attack and stroke. To explain heart injuries during gaming, several methods have been proposed. QT abnormalities on an EEG are the principal indicators of a heart problem and are triggered by oxidative damage of the heart’s mitochondria, associated with other factors, such as elevated SBP. Aggravating and predisposing factors also include poor sleep, obesity or metabolic disorders, history of seizures triggered by flashing lights, and prolonged sitting posture in social isolation. However, while there are some investigations on the causes of fatalities in PEGs, most have been performed post-mortem, and there are very few studies that have studied the heart and the hemodynamic in vivo during intensive gaming in PEGs. The expected increase in professional Esports triggers the importance of heart screening and appropriate physical training to prevent injuries. Albeit some recommendations exist on physical training [6], we did not find any study that compared different physical training regimes in professional Esports players, and this could be a direction for future studies. The social impact of injuries in Esports is expected to grow in future years, which might require a public health intervention to mitigate the occurrence of catastrophic events.

## 5. Conclusions

A limitation of the present review is that it includes a limited number of papers considering very high level Esports players and that only English literature was accessed. The retrieval of studies in the literature could have been more precise using more keywords and including literature in different languages. There are relatively few professional Esports players and the model of sport performance in Esports is very broad, depending on the type of games, thus, making it difficult to compare different researches. The growing number of PEGs fosters the need for scientific information to explain and prevent serious injuries, cardiovascular events, and sudden cardiac deaths during Esports. However, the conclusion that intensive competitive Esports can be fatal cannot be drawn directly, even though negative effects on health are clearly demonstrated. PEGs must be made aware that they are at increased health risk, and consequently, they should adopt healthy lifestyles and appropriate training. Heart rate variability, the presence of a clinical history of seizures, headaches, syncope, and angina pectoris should be carefully considered if found in the clinical record of Esports players. In the literature, there is a lack of information on different protocols for physical training that can prevent injuries. A drawback of this review is that it was difficult, in some paper, to differentiate between heavy Esports players and PEGs, because of the blurred boundaries between the two. This review further evidenced the need for new experimental studies in professional Esports to investigate the association of exertion with life threatening events, especially concerning the cardiac function. While traditional sports (except for extreme sports) over the years have been regulated in terms of intensity and duration, this is not the case for Esports, which do not have a history and are subject to changes in intensity and duration until the extremes.

Extreme effort in Esports is also a paradigm for the whole extreme sports sector. The life-threatening activities that have emerged in recent years impose serious challenges on society, and the public are encouraged to consider their high social costs. The findings from this review highlight a concern for public health and whether or not professional e-tournaments should be, in some way, regulated regarding duration and intensity to preserve the health of the players and avoid catastrophic events. Further research is necessary to elucidate the connection between intensity of effort and early signs of injury in Esports players in order to prevent life-threatening events during gaming.

## Figures and Tables

**Figure 1 ijerph-20-01968-f001:**
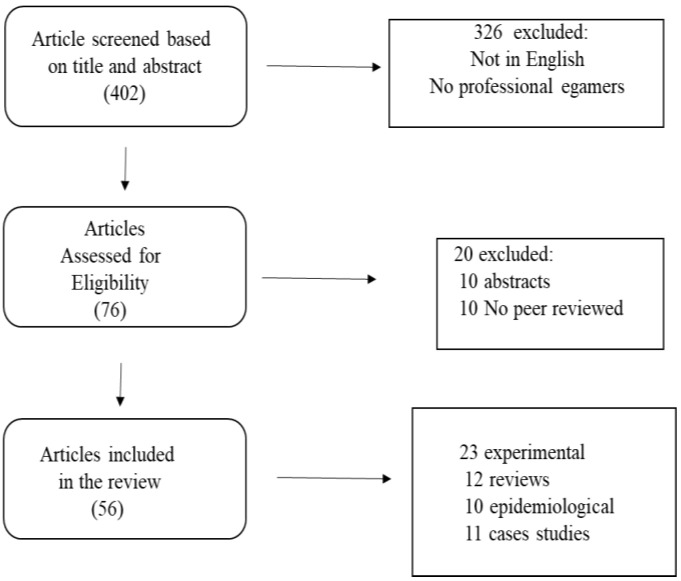
Flow chart of the literature selection.

**Figure 2 ijerph-20-01968-f002:**
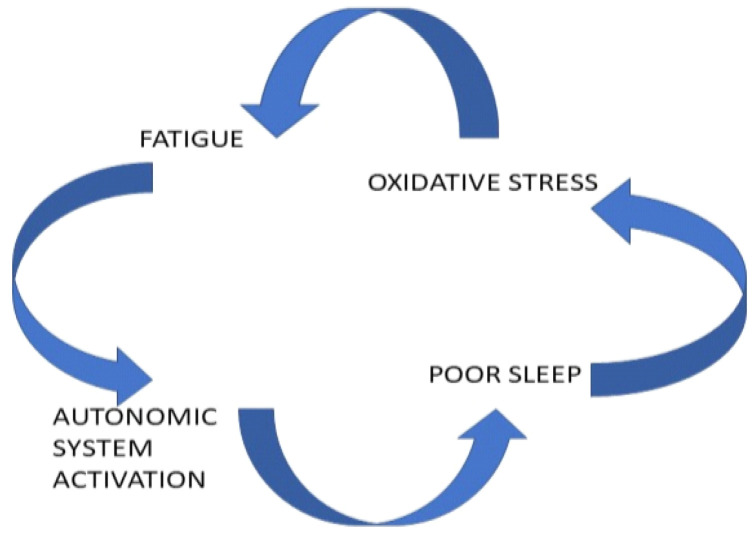
Interactions among the triggering factors in Esports injuries.

**Table 1 ijerph-20-01968-t001:** Studies on professional and semiprofessional e-gamers.

Reference	Study Type	Participants	Exposure	Key Findings
Lee, 2004 [13]	Case study	1 male, semiprofessional, 24 years old	4 days of playing with minimal sleep	Death
				Pulmonary arteries
				Complete thromboemboli
Chuang, 2006 [14]	Experimental	9 males, 1 female, professional, 14–30 years old	1–12 h/day	Seizures
				7 Subjects showed irregular EEG patterns at rest
Sainz et al., 2020 [15]	Experimental	15 males, professional, 22 years old	10.2 h/day	Increased reaction times after caffeine ingestion
Valladao et al., 2020 [16]	Experimental	23 males, semiprofessional, 21 years old	3 h match	Seated heart rate 120 ± 16 bpm vs. 81 ± 11 bpm at rest
Watanabe et al., 2021 [17]	Experimental	9 males, professional, 30.7 years old	Up to 3 h of gaming	Bpm 79 rest to 97 in gaming
				RMS of EMG of forearm increased
				Adaptation of HR to the opponent and to gaming conditions
Lee et al., 2021 [18]	Questionnaire	17 males, professional, 20 years old		6.8 h/night sleep
				86.4% sleep efficiency
Kang et al., 2020 [19]	Experimental	55 males, professional, 21.3 years old	Reaction times	Increased reaction times 1.0 vs. 8.3 ± 2.1 in controls
Song et al., 2007 [20]	Case study	1 male, professional, 40 years old	3 days continuous gaming	Rhabdomiolysis

## Data Availability

Not applicable.

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
