# Peer review of "Health Consequences of Intensive E-Gaming: A Systematic Review"

_ijerph, 2023, doi:10.3390/ijerph20031968_

Round 1
Reviewer 1 Report (Previous Reviewer 1)
The revisions are appropriate for the changes I wanted to see.
Author Response
Thank you very much for revising the paper and for the precious suggestions which helped to improve it. I revised the English accordingly to your suggestions.
Reviewer 2 Report (Previous Reviewer 3)
The paper is really improved.
Just pay attention to lines 25-28 (please do not repeat "respondents") and lines 65-68 (correct the sentece).
Thanks
Author Response
Thank you very much for reviewing the paper, I corrected accordingly to your suggestions..
Reviewer 3 Report (New Reviewer)
I found the manuscript to be very well written with no need for revision. The topic is a timely one as e-sport is becoming increasingly accepted with some US universities even offering scholarships to e-sport players. This manuscript importantly contributes to the increasing concern that e-sports need to be better supervised.
Author Response
Thank you very much for revising the paper.
I appreciate the problem is recognized and maybe this little work can contribute to rise the awareness on the problem of competitive egamers health.
This manuscript is a resubmission of an earlier submission. The following is a list of the peer review reports and author responses from that submission.
Round 1
Reviewer 1 Report
Line 10: write out professional esports players and then provide abbreviation
Line 47: write out APA before providing abbreviation
The abstract states that most deaths in PEGs are in males, but the body of the paper did not address this. Please elaborate on this statement in the paper… it could be as simply as most PEGs are men and only a small percentage is females; however, some type of follow-up is needed.
Author Response
Thank you, enclosed the answers

Reviewer 2 Report
Overall writing and terminology could still be improved, but the changes noted were mostly correct, but listed as comments and still need to be changed in the actual document in a few places. Otherwise it is quite close to being ready.
Author Response
Thank you, enclosed the response

Reviewer 3 Report
There are still strong mistakes in several senteces and these do not allow a proper reading.
Keep in mind that English is a language per se and not a translation of Italian grammar rules.
Remarkable is your effort in going through the subject.

Author Response
Thank you. Enclosed the answer

Reviewer 4 Report
I had the opportunity to review the paper entitled "Health Consequenses of Intensive EGaming. A systematic Review" .
The paper describes the consequenses of egaming to health in young people, but little experimental evidences have been described so yet. Furthermore, there is a plethora of reviews at the field.
The paper is in general well-written. However, english language changes are required.
Author Response
Thank you very much. Enclosed my answers.

Round 2
Reviewer 3 Report
Thanks for the adjustments.
Now the paper is improved, however I reccomend to express the sentences in lines 310-316 better (i.e."studies" is repeated too often).
Thanks
Reviewer 4 Report
None